# Effects of Virtual Reality Cognitive Training on Neuroplasticity: A Quasi-Randomized Clinical Trial in Patients with Stroke

**DOI:** 10.3390/biomedicines11123225

**Published:** 2023-12-06

**Authors:** Antonio Gangemi, Rosaria De Luca, Rosa Angela Fabio, Paola Lauria, Carmela Rifici, Patrizia Pollicino, Angela Marra, Antonella Olivo, Angelo Quartarone, Rocco Salvatore Calabrò

**Affiliations:** 1IRCCS Centro Neurolesi “Bonino-Pulejo”, S.S. 113, Cda Casazza, 98124 Messina, Italy; antonio.gangemi@irccsme.it (A.G.); rosaria.deluca@irccsme.it (R.D.L.); paola.lauria@irccsme.it (P.L.); carmela.rifici@irccsme.it (C.R.); patrizia.pollicino@irccsme.it (P.P.); angela.marra@irccsme.it (A.M.); antonia.olivo@irccsme.it (A.O.); angelo.quartarone@irccsme.it (A.Q.); 2Department of Economics, University of Messina, Via Consolare Valeria, 98125 Messina, Italy; rafabio@unime.it

**Keywords:** ischemic stroke, virtual reality, electroencephalography, neural plasticity, cognitive rehabilitation, brain injury, outcome measures, clinical trial, neurological disorders

## Abstract

Cognitive Rehabilitation (CR) is a therapeutic approach designed to improve cognitive functioning after a brain injury, including stroke. Two major categories of techniques, namely traditional and advanced (including virtual reality—VR), are widely used in CR for patients with various neurological disorders. More objective outcome measures are needed to better investigate cognitive recovery after a stroke. In the last ten years, the application of electroencephalography (EEG) as a non-invasive and portable neuroimaging method has been explored to extract the hallmarks of neuroplasticity induced by VR rehabilitation approaches, particularly within the chronic stroke population. The aim of this study is to investigate the neurophysiological effects of CR conducted in a virtual environment using the VRRS device. Thirty patients with moderate-to-severe ischemic stroke in the chronic phase (at least 6 months after the event), with a mean age of 58.13 (±8.33) for the experimental group and 57.33 (±11.06) for the control group, were enrolled. They were divided into two groups: an experimental group and a control group, receiving neurocognitive stimulation using VR and the same amount of conventional neurorehabilitation, respectively. To study neuroplasticity changes after the training, we focused on the power band spectra of theta, alpha, and beta EEG rhythms in both groups. We observed that when VR technology was employed to amplify the effects of treatments on cognitive recovery, significant EEG-related neural improvements were detected in the primary motor circuit in terms of power spectral density and time-frequency domains. Indeed, EEG analysis suggested that VR resulted in a significant increase in both the alpha band power in the occipital areas and the beta band power in the frontal areas, while no significant variations were observed in the theta band power. Our data suggest the potential effectiveness of a VR-based rehabilitation approach in promoting neuroplastic changes even in the chronic phase of ischemic stroke.

## 1. Introduction

Global epidemiological data indicate that approximately 16.9 million people suffer a stroke each year, resulting in a global incidence rate of 258 per 100,000 individuals, with variations between high- and low-income countries [1]. Survivors of stroke face immediate challenges in coping with long-term sensory-motor and cognitive impairments [1,2]. Deficits in language and communication, attention, visuo-spatial processing, long-term and procedural memory, reasoning and problem solving, as well as executive functions, are often present following a stroke.

Motor and cognitive alterations are the primary targets of neurorehabilitation, as they hinder the patients’ ability to perform activities of daily living (ADL) [3], and can lead to long-term medical issues (e.g., urinary incontinence), musculoskeletal problems (e.g., spasticity), and psychosocial complications (e.g., depression, emotional lability) [4]. Assessment of stroke impairment is fundamental to predict prognosis and functional recovery. To this aim, the NIH stroke scale [5,6], Barthel index [7], and FIM [8] are commonly used in clinical practice, although they lack a specific assessment of cognitive function. Moreover, evidence shows the validity and reliability of the modified Rankin scale (mRS) as a valuable instrument for assessing the impact of new stroke treatments [9]. Indeed, early, coordinated, and multidisciplinary rehabilitation plays a crucial role in promoting motor and cognitive recovery after stroke. Conventional stroke rehabilitation methods mainly involve physical therapy, occupational therapy, cognitive training, and speech therapy [10,11,12]. However, many stroke survivors still experience functional disabilities, with regard to cognitive deficits, which hinder their ability to perform daily activities.

Cognitive Rehabilitation (CR) is a therapeutic approach designed to improve cognitive functioning after a brain injury, including stroke, as well as in patients with neurodegenerative disorders. CR is a set of methods used to overcome cognitive deficits caused by stroke. Indeed, it includes different interventions aimed at improving the ability to perform cognitive tasks, achieved through the retraining of previously learned skills and/or the teaching of compensatory strategies. Two major categories of techniques, i.e., traditional and advanced, are widely used in the CR of patients with different neurological disorders [13]. Traditional techniques involve the use of cognitive strategies to retrain or alleviate deficits in the different cognitive domains by using a paper-and-pencil approach. On the other hand, advanced methods, including computer-assisted cognitive rehabilitation, use multimedia and informatics resources to potentiate neurocognitive performance [14].

In recent years, technology-based stroke rehabilitation interventions have shown promise in improving the motor and cognitive abilities and autonomy of stroke patients [15,16]. Advanced technology is increasingly being incorporated into stroke neurorehabilitation to enhance standard treatments, reduce neurological disability, and improve overall functioning. In fact, the integration of psychology, technology, and neuroscience allows for a better understanding of how virtual reality (VR) affects the cognition of the human brain [17]. VR is a commonly used advanced neurorehabilitation technology that aims to improve motor and cognitive abilities in stroke patients [18,19]. VR involves computer-based simulations that allow users to interact with multisensory environments, receiving real-time feedback on their performance [20,21]. This technology promotes repetitive and task-specific training, actively involving patients, providing constructive feedback, and accurately measuring functional improvement. Recent studies have shown the efficacy of cognitive and motor rehabilitation through the use of virtual environments, such as the VRRS Evo-4 machine, where patients interact with common images with physical properties simulated by the tool, allowing exercises designed to stimulate the main cognitive domains.

The assessment of cognitive outcomes is often performed using clinical tests (such as the Oxford Cognitive Screen), which are standardized and validated within this patient population [22]. However, they are user-dependent, and scores are influenced by the motivation and collaboration of the patient. Therefore, more objective tools to better investigate functional outcomes are needed for both research and clinical purposes.

Electrophysiological approaches, such as EEG, offer a high temporal resolution in the order of milliseconds and can serve as a set of biomarkers crucial for assessing cognitive changes observed during various activities and challenging conditions, including VR [23,24,25,26]. Studies on neurophysiological changes associated with VR neurorehabilitation are relatively new, with initial evidence from behavioral tasks in healthy individuals [27,28]. Currently, EEG is utilized in VR therapy to monitor and provide augmented feedback regarding cortical activation [29,30] during motor and cognitive tasks [31,32,33].

Therefore, our purpose is to investigate the neurophysiological effects of cognitive stimulation training conducted in a virtual environment using the VRRS device on patients in the chronic phase of ischemic stroke. We are particularly interested in evaluating how this advanced training impacts brain plasticity mechanisms. To this aim, we directly measured brain electrical activity through EEG recordings.

Through this investigation, we sought to gain valuable insights into the potential of VR cognitive stimulation as a neurorehabilitation approach for patients with neurological disorders and contribute to the understanding of the underlying neural mechanisms involved in the observed cognitive improvements. The findings from this study could have implications for the development of innovative and effective interventions to enhance cognitive recovery in individuals with stroke or other neurological events.

## 2. Materials and Methods

### 2.1. Study Setting and Participants

Thirty—moderate-to-severe patients with ischemic stroke in the chronic phase (at least 6 months after the event), with a mean age of 58.13 (±8.33) for the experimental group and 57.33 (±11.06) for the control group, were enrolled in this study. They attended the outpatient clinic of the Neurorehabilitation Unit of IRCCS Neurolesi “Bonino Pulejo” (Messina, Italy) from October 2022 to March 2023. A more detailed description of the two groups is provided in Table 1.

The stroke patients and/or their family members were provided with adequate information about the study and offered the opportunity to participate with written consent. The study adhered to the principles set forth in the Declaration of Helsinki on Human Rights, and the local Ethics Committee approved the study (IRCCS-ME-CE 08/21). Inclusion criteria were as follows: (1) diagnosis of first right ischemic stroke in the chronic phase, i.e., ≥ 6 months after the event; (2) age range 18–75; and (3) absence of disabling sensory impairment (i.e., hearing and visual impairment); (4) Rankin Scale score ≥ 3; (5) Barthel index ≥ 5. Exclusion criteria were as follows: (1) intake of psychoactive drugs potentially interfering with the training; (2) presence of neurological disorders other than the first ever ischemic stroke; and (3) absence of the ability to understand verbal delivery of a simple order, Token Test ≤ 4; (4) presence of debilitating behavioral alterations and severe psychiatric symptoms.

### 2.2. Procedures

Thirty patients were randomly assigned to one of 2 groups, with fifteen allocated to the experimental group (EG) and the other fifteen to receive standard treatments, forming the control group (CG), based on the order of recruitment (in order to meet the criteria for a quasi-randomized study). We used the sample size calculator, a public service of Creative Research Systems survey software (https://www.surveysystem.com/sscalc.htm), to determine an adequate and minimal sample size to exclude systematic error, established with a confidence level of 95% and a confidence interval of 2%.

The experimental group received neurocognitive stimulation using virtual reality training (VRT) using the Virtual reality rehabilitation system (VRRS), while the control group received the same amount of standard neurorehabilitation (using a paper-and-pencil approach). To study neuroplasticity changes (that was the aim of the study), we focused on investigating the power band spectra of theta, alpha and beta EEG rhythms in both groups. Theta and alpha rhythms are specific frequency bands of brain electrical activity that have been associated with various cognitive processes and neural plasticity [34].

By comparing the changes in theta and alpha EEG rhythms between the experimental group (who received the VR cognitive stimulation) and the control group (who did not receive this advanced stimulation during the study period), we aimed to assess the specific effects of VR on brain plasticity in patients with chronic stroke.

EEG data were recorded by the neurophysiology technical staff under the supervision of the neurological physician and acquired using a gold-standard digital EEG amplifier (Micromed Medical System, Treviso, Italy). The system continuously recorded EEG signals on 19 channels. Electrodes were placed on the patient’s scalp according to the International Measurement System 10/20 criteria, with the reference electrode positioned in the ear and the ground electrode placed posteriorly at Fz. The preprocessing of the EEG was performed in Matlab using EEGLAB [35].

First, the baseline was removed from each channel. Then, high EEG signals were filtered at 0.5 Hz to remove respiratory noises, while low signals were filtered at a cutoff frequency of 50 Hz to eliminate high-frequency noises. A notch filter was also applied at 50 Hz to remove power line interference. The signals were visually inspected to manually remove residual artifacts. The data were then segmented into epochs of 4 s free of artifacts.

For the analysis of brain rhythms, data were recorded from the occipital areas for alpha rhythm and from the fronto-temporal areas for beta and theta rhythms bilaterally, although the focus of our research was primarily aimed at investigating the neuroplastic effects in the right-lesioned hemisphere. Electrodes were placed across the entire scalp to ensure comprehensive coverage [36].

EEG data were recorded during a 20 min session where the patient was at psychosensory rest, with their eyes closed. For short periods, the patient was asked to open their eyes to assess the alpha rhythm reactivity recorded in the occipital cortex areas. Quantitative analysis was performed using custom algorithms developed in Matlab code. The power spectral density (PSD) was evaluated by transforming the signal from the time domain to the frequency domain using the Welch method [37]. PSDs were calculated for each epoch and then averaged. The absolute total power of the signal and absolute power of each band were computed for each electrode. The bands considered were theta (4–7 Hz), alpha (8–13 Hz), and beta (14–29 Hz).

### 2.3. Virtual Cognitive Task Using VRRS

VRRS is an advanced rehabilitation platform designed to facilitate the recovery process for patients with neurological conditions. It is one of the most comprehensive and clinically tested virtual reality systems for rehabilitation and tele-rehabilitation. The VRRS utilizes an exclusive magnetic kinematic acquisition system and offers a range of rehabilitative modules, including neurological, logopedic, and cognitive tasks, catering to a wide spectrum of neurological diseases. The system incorporates augmented feedback to enhance physiological learning, providing patients with specific information on their movements to improve the quality of their performance. Each exercise is accompanied by a preview that demonstrates to therapists and patients how the sensors should be positioned and the correct way to perform the exercise (Figure 1).

During the EEG execution, the enrolled patient sits in front of the VRRS device, actively engaging with it to perform the virtual cognitive task.

The VRRS cognitive module comprises a set of interactive activities designed for specific cognitive domains. These activities include tasks related to attention and memory, such as selective and sustained attention, as well as verbal or visual-spatial activities. The virtual tasks provided through the VRRS can be categorized into two main types based on the method of interaction with the virtual reality tool. The first category encompasses 2D exercises where the patient interacts with objects and scenarios through the touch screen or a specialized magnetic tracking sensor coupled with a squeezable object, effectively emulating mouse-like interaction capabilities. The second category involves 3D exercises, allowing patients to interact with virtual scenarios and objects using magnetic wearable sensors typically placed over the hand, enabling 3D position tracking of the end effector. This 3D modality allows for movements of the upper and lower limbs in three dimensions while interacting with the virtual environment (see Table 2). During the EEG examination, the virtual task is administered with three levels of difficulty for execution time, and the number of stimuli-targets and distractors presented to the patient is controlled (see Table 2) [38].

### 2.4. Standard Cognitive Training

The CG performed a conventional cognitive rehabilitation program with the same features and amount of time/intensity as the experimental one. However, cognitive domains were stimulated by using the classical paper-and-pencil approach instead of using VR (see Table 2). In fact, the cognitive-oriented intervention administered to CG included a series of standard face-to-face activities, organized for specific cognitive domains (attention processes and memory abilities) and relative cognitive sub-domains (selective and sustained attention processes, verbal and visuo-spatial memory), without the use of virtual systems.

### 2.5. Statistical Analysis

Data analysis was performed using IBM SPSS Statistics, Version 24 (IBM, Armonk, NY, USA) [39]. A mixed-model ANOVA for repeated measures was applied, with three repeated factors (bands: alpha, beta, and theta bands) and time (T0—pre-intervention baseline, T1—post-test) and one between subject factor (group: experimental group with VRT and control group without VRT). A Bonferroni correction was applied for multiple comparisons. The alpha level was set to *p* < 0.05 for all statistical tests. In the case of significant effects, the effect size of the test was reported. The effect sizes were computed and categorized according to eta squared (η^2^).

## 3. Results

Regarding the demographic statistics, Table 3 shows the means and standard deviations for the provided demographics and indices. This table presents a comprehensive summary of the key characteristics of both the experimental and control groups, including age, years of education, Barthel index, Rankin Scale score, years from ischemic stroke, and sex distribution. To ensure the balance between the groups, we conducted a thorough analysis of the provided demographic and clinical characteristics. As shown in Table 3, we assessed the distribution of sex, age, years of education, Barthel index, Rankin Scale score, and years from ischemic stroke between the two groups. Independent samples *t*-tests for continuous variables and chi-squared tests for categorical variables were applied to compare the two groups. The statistical tests indicate that there are no significant differences between the groups for any of these variables. Therefore, we can confidently conclude that the groups are well-balanced in terms of these important baseline characteristics.

Concerning the right hemisphere, the mixed-model ANOVA for repeated measures revealed a significant effect of the factor “Group” (F (1, 28) = 4.11, *p* < 0.05, η^2^ = 0.09), indicating a difference between the experimental and control groups.

The “Group × Time” interaction showed significant effects (F (1, 28) = 11.03, *p* < 0.01, η^2^ = 0.11), suggesting that neurocognitive stimulation using VRT resulted in significant differences between the experimental and control groups in the alpha and beta bands. Furthermore, the “Group × Bands × Time” interaction exhibited significant effects (F (2, 56) = 78.45, *p* < 0.01, η^2^ = 0.12), indicating that the alpha and beta bands showed an increase from pre-test to post-test for the experimental group, respectively, t (29) = 4.92, *p* < 0.01, d = 0.88 and t (29) = 6.01, *p* < 0.01, d = 0.91, while the theta band exhibited no increase, t (27) = 0.46, *p* = 0.35 (see Figure 1 and Figure 2). With reference to the control group, no differences in the single bands were observed. To summarize, VR enhanced only the alpha and beta frequency bands (Table 4).

Regarding the left hemisphere, a significant effect of the factor “Group” (F (1, 28) = 5.93, *p* < 0.05, η^2^ = 0.11) emerged, indicating a difference between the experimental and control groups within the left hemisphere.

The “Group × Time” interaction showed significant effects (F (1, 28) = 5.03, *p* < 0.05, η^2^ = 0.11), suggesting that neurocognitive stimulation using VRT resulted in significant differences between the experimental and control groups in the alpha and beta bands. Furthermore, the “Group × Bands × Time” interaction exhibited significant effects (F (2, 56) = 8.19, *p* < 0.01, η^2^ = 0.09), indicating that the alpha and beta bands showed an increase from pre-test to post-test for the experimental group, respectively, t (29) = 5.12, *p* < 0.01, d = 0.90; and t (29) = 5.87, *p* < 0.01, d = 0.91, while the theta band exhibited no increase, t (27) = 0.38, *p* = 0.47 (see picture n. 2 and picture n. 3). With reference to the control group, no differences in the single bands were observed. To summarize, VR enhanced only the alpha and beta frequency bands (Table 4).

In summary, the results suggest that neurocognitive stimulation through VRT had a significant impact on the alpha and beta frequency bands. These bands exhibited increased activity in the experimental group, while the control group showed no significant changes in any of the bands. This indicates that VRT enhances cognitive functioning primarily in the alpha and beta frequency bands.

## 4. Discussion

The study of EEG signals, correlated with their application in different technologies, holds great interest in neuroscience, particularly in the fields of assistive technology and neurorehabilitation, where the external stimulus can be provided through VR [40]. In a previous study, we investigated robotic-based rehabilitation (using the Lokomat, Hokoma, Zurigo) combined with VR in patients with chronic hemiparesis. We found that improvement in gait and balance was paralleled by important EEG signal modifications. In particular, the EEG data suggested that the use of VR may entrain several brain areas involved in motor planning and learning, thus leading to an enhanced motor performance [41].

In the current study, we confirmed that VR training could be effective in improving neuroplasticity (and potentially cognitive functioning) in patients in the chronic phase of ischemic stroke, as also demonstrated by other authors in different types of neurological disorders [42,43,44,45,46].

However, as far as we know, this is the first study focusing on the importance of assessment of electrophysiological changes after a brain injury involving only the right hemisphere. Indeed, our training and investigation mainly concerned visual working memory alterations, deficits of the speed of response and sustained attention, which are often underestimated signs of right hemisphere injury, despite their significant impact on rehabilitation outcomes [47,48]. This important role of the right hemisphere has also been demonstrated by neuroimaging studies [49,50,51,52,53].

The present study aimed to observe the effects of an innovative neurocognitive stimulation VR system, using training based on specific attention and memory abilities, on the EEG bands of two groups of subjects in the chronic phase of ischemic stroke. The findings revealed a significant increase in both alpha and beta bands in the experimental group following the intervention, suggesting that VRT had a positive impact on neural oscillatory activity, especially in the right hemisphere (which was the focus of the present study). The observed increase in alpha and beta bands aligns with previous research, demonstrating the neuroplasticity-inducing effects of VR-based interventions on brain activity [54,55,56,57,58,59,60,61]. The ability of VRT to engage and challenge the neural networks involved in cognitive processes may account for the observed changes in neural oscillations [62]. Specifically, the increase in alpha power could indicate enhanced attentional focus and cognitive resource allocation [29,56], while the rise in beta power may reflect improved cognitive control and motor planning [63]. The control group did not exhibit significant changes in alpha and beta bands over time. This suggests that the observed neurocognitive effects were specific to the VRT intervention and not simply a result of the passage of time or other non-specific factors.

These findings hold implications for the use of VRT as a potential tool in cognitive rehabilitation and neurocognitive enhancement. By understanding the neural mechanisms underlying the effects of VRT, future research can design targeted interventions to optimize cognitive training programs and potentially improve outcomes for individuals with cognitive deficits or neurodevelopmental disorders.

In addition to the implications for CR, the findings of this study could have broader implications for the field of neurorehabilitation and brain plasticity research. Virtual reality technology has shown promise as a versatile and effective tool for promoting neural plasticity in various clinical populations. By providing an immersive and engaging environment, VRT can target specific cognitive domains and facilitate neurocognitive, as well as motor reorganization. It has been shown that VR, due to the use of auditory and visual feedback, may affect different perceptual and experiential aspects. This complex sensory stimulation may increase the patient’s awareness of his/her results (knowledge of results) as well as awareness of performance (knowledge of performance), inducing changes in neural plasticity processes with a consequent reinforcement of learning [20,45]. These positive effects on cortical plasticity could be due to the reactivation/amplification of brain neurotransmission within spared or unused circuits. It is also conceivably due to the involvement of mirror neurons, facilitated by the visual-motor information coming from the observation of the stimuli on the VR screen [24,25,26,27,28,29,30,31,32,33,34,35,36,37,38,39,40,41,42,43,44,45]. Moreover, by using VR environments, it is possible to perform tasks that may be too difficult, time-consuming, or impossible to perform in a natural world setting. It is noteworthy that VR enables healthcare professionals to provide standardized rehabilitation protocols, controlled stimulus presentations, and clinical progress and performance measures [45]. These outcome measures may become more objective if investigated (as in our study) with electrophysiological tools. Then, the advanced training can be tailored to the patient’s clinical status and needs, also providing personalized feedback on performance.

Nonetheless, among the main concerns regarding the use of innovative technology, including VR, are system usability and the high costs. Most of the non-immersive VR devices, such as the VRRS, are easy to use and do not always require the presence of a caregiver to set the device if the patient does not have severe cognitive-behavioral problems [64]. On the contrary, semi-immersive and immersive devices (except for the Oculus) need a therapist to properly use the tool and supervise the training. As regards the costs, most VR devices have lower costs than other innovative rehabilitation treatments (e.g., robot-assisted training), and can be used at home if a telemedicine service is available.

## 5. Limitations of the Study and Future Perspectives

This pilot study has some limitations. First, the small sample size may prevent generalizing the results to the entire stroke population. However, the sample is homogeneous, as it is composed of patients with chronic right ischemic stroke. Moreover, to strengthen the generalizability of the results to neurorehabilitation, larger sample sizes and more diverse populations should be included.

Second, a quasi-randomization method may lead to selection bias. Therefore, randomized clinical trials are needed to confirm these promising results.

Third, we did not assess any behavioral tools after the intervention, so we cannot determine if and to what extent VR may have influenced our patients’ clinical outcomes. Nonetheless, there is a lack of literature demonstrating that patients undergoing VR have superior (or at least the same) results than conventional training [65,66,67,68,69]. Also, this issue was outside the scope of the article, which focused on demonstrating the underpinning of VR-inducing neuroplasticity. The specific VRT protocols and tasks used in this study may have influenced the observed outcomes, and further research with a broader range of VR paradigms is warranted. Finally, although the present study focused on evaluating the power in the alpha, beta, and theta frequency bands in both hemispheres [70], we focused on the affected right hemisphere, and we did not calculate interhemispheric asymmetry. The latter would have definitely provided more detailed data on the functional connectivity (and therefore functional recovery) between the two cerebral hemispheres. However, the analysis of frequency band power is an approach also used in other studies to examine the intensity of brain waves in different regions or frequency bands [70]. This analysis can be conducted on data from individual hemispheres or specific regions without considering the symmetry parameter between the two hemispheres. In future research, we plan to integrate a comprehensive neurophysiological assessment of symmetry and coherence parameters along with hemodynamic studies using fNIRS to achieve improved spatiotemporal resolution.

As technology continues to advance, and VR platforms become more sophisticated, the potential of VRT for neurorehabilitation and cognitive enhancement will likely expand. Integrating VRT with other rehabilitative approaches, such as traditional therapy or brain stimulation techniques, could lead to synergistic effects and further optimize patient outcomes. Finally, by harnessing the potential of VRT as a tool for neurocognitive stimulation and brain plasticity, healthcare providers may be better equipped to design tailored and effective interventions for individuals with diverse cognitive challenges, including those with stroke, traumatic brain injury, and neurodevelopmental disorders.

## 6. Conclusions

In conclusion, this study may contribute valuable insights into the effects of neurocognitive stimulation using virtual reality training on brain oscillatory activity in the chronic phase of ischemic stroke. The significant increase in alpha and beta bands observed in the experimental group highlights the potential of VRT to induce neuroplastic changes in the brain. By providing an immersive and engaging environment, VRT can target specific cognitive domains and facilitate neurocognitive as well as motor reorganization. Through this investigation, we sought to contribute to the understanding of the underlying neural mechanisms involved in cognitive recovery after innovative approaches like virtual reality. In fact, the findings from this study could have implications for the development of innovative and effective interventions to enhance cognitive function in individuals with stroke, as well as other neurological disorders.

This research sets the stage for future investigations that can refine VRT protocols and pave the way for innovative and personalized cognitive rehabilitation interventions, ultimately benefiting individuals striving to overcome cognitive impairments and improve their attention processing and memory abilities. Indeed, larger multicenter RCTs are needed to confirm these promising findings and to investigate the role of other EEG bands as well as interhemispheric EEG coherence in the functional recovery following a brain injury.

## Figures and Tables

**Figure 1 biomedicines-11-03225-f001:**
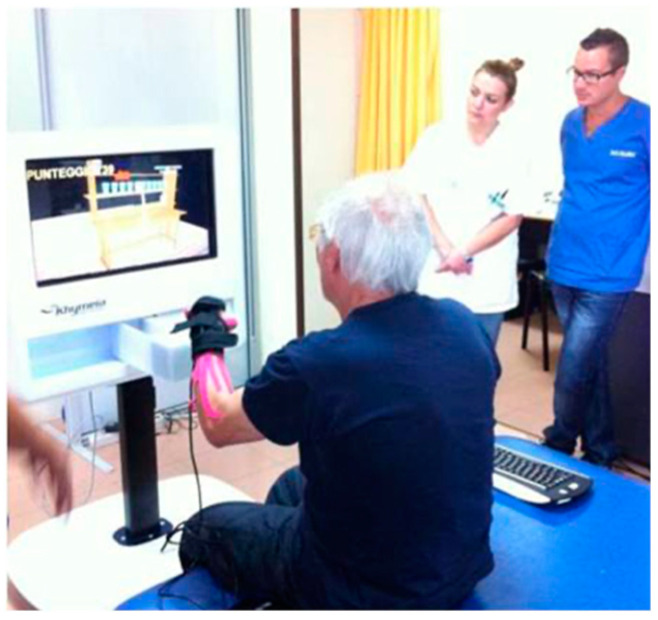
A patient supervised by two clinicians during the execution of a motor-cognitive exercise to improve sustained attention.

**Figure 2 biomedicines-11-03225-f002:**
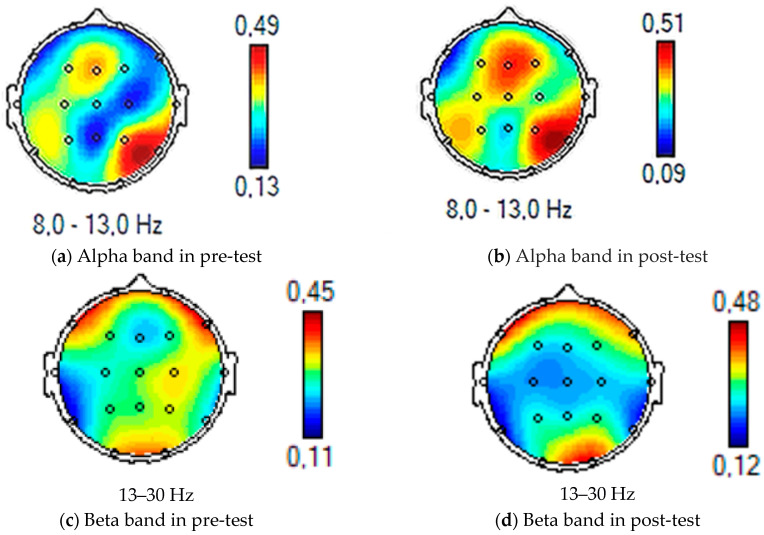
The power spectral density in the alpha and beta bands of a subject randomly selected from a single case in the experimental group in both the pre-test and post-test phases is shown in the figure.

**Table 1 biomedicines-11-03225-t001:** Demographic and clinical description of both the experimental and control samples at the beginning of the study.

Subject	Age (Years)	Gender	Education (Years)	Barthel Index	Rankin Scale	Time Elapsed Since the Event
Experimental Group						
1	57	M	8	15	5	6
2	61	M	8	5	5	8
3	69	M	5	40	4	7
4	65	M	13	15	4	6
5	63	F	13	10	4	12
6	57	F	8	10	4	6
7	57	F	13	35	4	8
8	39	M	8	70	3	8
9	59	M	8	10	5	7
10	56	F	8	40	4	6
11	60	F	8	65	3	7
12	43	M	13	10	5	9
13	67	M	13	5	5	9
14	66	M	13	10	5	12
15	53	M	13	25	4	6
Control Group	**Age (Years)**	**Gender**	**Education (Years)**	**Barthel Index**	**Rankin Scale**	**Time Elapsed Since the Event**
1	38	M	8	10	5	6
2	73	M	13	5	5	6
3	59	M	8	35	4	7
4	73	M	5	20	4	12
5	68	M	5	15	4	8
6	69	F	8	15	4	8
7	65	F	8	30	4	6
8	64	M	13	70	3	6
9	55	M	5	40	5	6
10	54	F	8	40	4	7
11	48	M	13	55	4	9
12	55	F	10	5	5	6
13	50	M	13	20	4	7
14	45	F	8	15	4	7
15	44	F	8	10	4	8

**Table 2 biomedicines-11-03225-t002:** Cognitive rehabilitation program: virtual reality task using VRRS—Evo and standard activities administered during EEG signal processing.

Domain	Sub-Domain	VRRS Task	Standard Activities
-Attention Processes	Selective	To administer the scanning exercise, the user must locate the target symbols in a grid and select the matching virtual symbols. To select and immediately recall feedback (audio and video) similar to various elements (colors, musical strings, geometric or abstract forms, animals, numbers) observed in the virtual environment, the patient touches the virtual target element within a specific time. This action causes a visual change with a specific audio feedback (positive reinforcement), using VVRS—interaction between the cognitive therapist and the patient. Otherwise, the element disappears (negative reinforcement).	To administer the attention exercise, the user must locate the target symbols while facing a paper-and-pencil grid and select the matching real symbols. To select and immediately recall feedback (audio and video) resembling various elements (colors, musical strings, geometric or abstract forms, animals, numbers) observed in the real environment, the patient touches the target element within a specific time, using a timer and the interaction between the cognitive therapist and the patient.
	Sustained	To stimulate sustained attention processes, the patient observes from 3 to 5 target stimuli for a variable and progressive time (10–15 min), with an attentional focus on the virtual tasks administered.	To stimulate sustained attention processes, the patient observes from 3 to 5 target stimuli for a variable and progressive time (10–15 min), with an attentional focus on the real activities administered.
Memory Abilities	Verbal	To work on recognition and remembrance in virtual tasks involving verbal material, reminiscence and validation therapy, mnemonic techniques, and strategic skills.	To work on recognition and remembrance in traditional tasks with paper-and-pencil verbal material, reminiscence and validation therapy, mnemonic techniques and strategic skills, face to face with a therapist, without a virtual tool.
	Visuo-Spatial	To work on recognition and remembrance virtual tasks with not verbal/visuo-spatial tasks (pictures; image; number; colors…) mnemonic techniques and strategic skills.	To work on recognition and remembrance using paper-and-pencil tasks without verbal/visuo-spatial tasks (pictures, images, numbers, colors), employing conventional mnemonic techniques and strategic skills, face to face with a therapist, without the use of virtual tools.

**Table 3 biomedicines-11-03225-t003:** Summary of the demographic and clinical description of both the experimental and control samples and statistical comparisons.

Socio-Demographic and Clinical Variables	Experimental Group	Control Group	Statistic	Pairwise Comparisons
Sex (male/female) ^a^	M = 10	M = 10	0.00	(*p* = 1)
	F = 5	F = 5		
Age (years) ^b^	58.13 (8.33)	57.33	0.24	*p* = 0.82
Education level (years) ^b^	10.13 (2.87)	8.96 (2.92)	1.19	*p* = 0.24
Barthel index (0–100) ^b^	24.33 (21.20)	25.66 (19.07)	0.18	*p* = 0.85
Rankin Scale score (0–6) ^b^	4.26 (0.70)	4.20 (0.56)	0.28	*p* = 0.77
Years from ischemic stroke ^b^	7.8 (2.00)	7.26 (1.62)	0.80	*p* = 0.43

^a ^Chi-square test (critical value). ^b^ Independent samples *t*-test: mean (standard deviation).

**Table 4 biomedicines-11-03225-t004:** Means and standard deviations of theta, alpha, and beta bands (microvolts) in the right and left hemispheres, and *p*-values.

	Pre-Test	*p*	Post-Test	*p*		
Right Hemisphere (Hz)	Experimental	Control		Experimental	Control	
** *Theta band* **	(M = 17.80; SD = 2.24)	(M = 18.30; SD = 1.73)	0.23	(M = 18.10; SD = 2.24)	(M = 18.02; SD = 1.76)	0.16
** *Alpha band* **	(M = 21.33; SD = 0.97)	(M = 21.41; SD = 1.02)	0.31	(M = 30.23; SD = 2.99)	(M = 21.8; SD = 1.02)	0.01
** *Beta band* **	(M = 23.13; SD = 2.74)	(M = 23.27; SD = 2.89)	0.37	(M = 28.27 SD = 2.37)	(M = 23.27; SD = 2.43)	0.01
**Left Hemisphere** ** *Theta band* **	(M = 17.40; SD = 2.74)	(M = 18.03; SD = 1.77)	0.22	(M = 18.25; SD = 2.24)	(M = 18.72; SD = 1.76)	0.14
** *Alpha band* **	(M = 22.43; SD = 1.67)	(M = 21.32; SD = 1.32)	0.36	(M = 30.23; SD = 2.99)	(M = 21.8; SD = 1.02)	0.01
** *Beta band* **	(M = 23.53; SD = 3.15)	(M = 23.40; SD = 2.47)	0.49	(M = 26.97 SD = 3.81)	(M = 23.13; SD = 2.90)	0.05

## Data Availability

Data will be available on request to the corresponding author.

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
