# Peer review of "Effects of Virtual Reality Cognitive Training on Neuroplasticity: A Quasi-Randomized Clinical Trial in Patients with Stroke"

_biomedicines, 2023, doi:10.3390/biomedicines11123225_

Round 1

Reviewer 1 Report

Comments and Suggestions for Authors

The problem of rehabilitation of patients in the chronic phase of ischemic stroke is relevant, despite the long period of its study around the world. Inter-hemispheric asymmetry introduces additional difficulties in the rehabilitation of patients who have suffered a right brain stroke, since such patients may have problems with speech comprehension, behavior, spatial gnosis, and compliance with treatment and cooperation with a doctor. The development of new methods of neurorehabilitation using new computer technologies, including virtual reality tasks, may be able to solve these problems. This explains that the main topic of this manuscript has clinical and scientific significance.

However, in its current form, the manuscript cannot be accepted for publication and needs very serious revision.

Abstract:

Present a structured abstract, including the purpose of the study, materials and methods, results, conclusion.

Introduction:

Replace the phrase "chronic stroke" with "chronic phase of ischemic stroke" wherever you use it, including the title, abstract and manuscript.

Material and Methods:

The sample size is very small, which casts doubt on the results obtained by the authors. What method was used by the authors to calculate the minimum sample size and exclude systematic error?

The authors note that the study involved 28 participants who were divided into two groups. However, the data in table 1 shows that there were 15 participants in each group (30 in total). What criteria were used when assigning participants to comparison groups? In addition, the age of patients in these small groups was very heterogeneous. For example, the age of the control group participants ranged from 38 to 72 years. This may affect the speed of virtual reality tasks.

How can the authors explain the inclusion in the study of only patients who have suffered a right brain stroke? Such localization of stroke is often accompanied not only by paralysis of the left side of the body, but also by vision problems, rapid, overly curious behavior, poor decision-making and cognitive disorders. How well did the observed patients understand the virtual tasks assigned to them?

What rehabilitation methods were used in the control group? What did the authors include in the concept of "standard rehabilitation" (lines 113-114)? Was pharmacotherapy used in the control and comparable groups? If "yes", what medications and what doses were prescribed to the participants in this study?

Line 144 - Please remove the abbreviation from the subsection name.

Results:

This section is the "weakest link" of the manuscript.

In the Materials and Methods section (lines 119-121), the authors indicated that they focused their attention on the study of the characteristics of theta and alpha rhythms of the EEG. However, the Results section presents the characteristics of beta and alpha rhythms. There is no information about the dynamics of changes in theta rhythm characteristics.

It is unclear in which lead(s) the authors analyzed the frequency of alpha and beta rhythms? Was monopolar monoarticular mounting used? Or was it an average montage? The frequency of the alpha band was analyzed in the occipital leads? Right or left? Over which hemisphere were the scalpel electrodes placed? Was the interhemispheric asymmetry of the average frequency of the EEG rhythms analyzed by the authors or was it not in the pre-test and post-test? Was the beta rhythm frequency also analyzed in the occipital leads (above the occipital electrodes) or in the frontal leads? Right or left? Etc.

In general, this section needs serious revision. I recommend the authors to present the results of statistical analysis of the characteristics of all the studied EEG frequency bands, including theta, alpha and beta (for examples, mean frequency, mean power, mean amplitude, coefficient of hemispheric asymmetry for all these characteristics) in the pre-test and post-test.

Discussion:

Lines 237-239 - the authors indicate that no positive EEG changes were found in the control group. Could this mean that "standard rehabilitation" is not effective? With what method of "standard rehabilitation" was the virtual reality test compared? Was "Standard rehabilitation" not prescribed to patients in the experimental group?

How does the virtual reality test method developed by the authors differ from similar numerous others previously known or actively studied? What is the novelty of this study?

Comments on the Quality of English Language

The manuscript needs a moderate revision of the special English language, primarily the style, including the use of some phrases and terms.

Author Response

However, in its current form, the manuscript cannot be accepted for publication and needs very serious revision. 

  • Abstract: 

Present a structured abstract, including the purpose of the study, materials and methods, results, conclusion. 

As per journal style and academic editor advice, abstract should be unstructured. However, we have revised it, following the paragraph suggested.  

  • Introduction: 

Replace the phrase "chronic stroke" with "chronic phase of ischemic stroke" wherever you use it, including the title, abstract and manuscript. 

We re-pleased it, as suggested.  

Material and Methods: 

The sample size is very small, which casts doubt on the results obtained by the authors. What method was used by the authors to calculate the minimum sample size and exclude systematic error?  

We have reported in the text the method used to calculate the minimum sample size and exclude systematic error, as requested. 

The authors note that the study involved 28 participants who were divided into two groups. However, the data in table 1 shows that there were 15 participants in each group (30 in total).  

There are thirty patients enrolled as shown in the descriptive table n.1. It was an error in the text that we have corrected.  

What criteria were used when assigning participants to comparison groups? In addition, the age of patients in these small groups was very heterogeneous. For example, the age of the control group participants ranged from 38 to 72 years. This may affect the speed of virtual reality tasks. 

The patients were randomly assigned to one of 2 groups (experimental group [EG], or standard treatment, namely the control group [CG]) in order of recruiting (that’s why we added in the title a quasi randomised trial). The criteria used to enroll in the control group are the same as those used for the experimental group. these criteria are reported in the text, both inclusion and exclusion criteria, including an age range beetween 18-75.  The average ages of the two groups do not differ significantly, (58,1 is a mean of age for an experimental group; and 57,3 for the control group).  

How can the authors explain the inclusion in the study of only patients who have suffered a right brain stroke? We have focused on right laterality, because patients with left lesions are more often aphasia with deficit of comprehension (which was an exclusion criteria). Moreover, patients with right lesions have more often attention and visuo-spatial problems, that were properly addressed by our training (both conventional and virtual). 

Such localization of stroke is often accompanied not only by paralysis of the left side of the body, but also by vision problems, rapid, overly curious behavior, poor decision-making and cognitive disorders.  

As better specified in inclusion criteria, patients with vision problems, or cognitive-behaviors alterations have been excluded. 

How well did the observed patients understand the virtual tasks assigned to them? 

All the patients observed were able to have preserved the understanding of the verbal delivery of simple order, with a Token Test ≥ 4. We have reported in the exclusion criteria.  

What rehabilitation methods were used in the control group? What did the authors include in the concept of "standard rehabilitation" (lines 113-114)? 

We have described the methods used in the control group and explained the concept of standard rehabilitation, as suggest.  

Was pharmacotherapy used in the control and comparable groups? If "yes", what medications and what doses were prescribed to the participants in this study? The use of drugs potentially interfering with the cognitive status was an exclusion criteria in both groups, as specified. 

Line 144 - Please remove the abbreviation from the subsection name. 

Done 

Results: 

This section is the "weakest link" of the manuscript. 

In the Materials and Methods section (lines 119-121), the authors indicated that they focused their attention on the study of the characteristics of theta and alpha rhythms of the EEG. However, the Results section presents the characteristics of beta and alpha rhythms. There is no information about the dynamics of changes in theta rhythm characteristics.  

Mismatch Between Materials and Methods and Results: You've pointed out a discrepancy between our stated focus on theta and alpha rhythms in the Materials and Methods section and the presentation of characteristics of beta and alpha rhythms in the Results section. We acknowledge this oversight, and we apologize for any confusion it may have caused. We have incorporated the theta results into our finding (highlighted in green). 

It is unclear in which lead(s) the authors analyzed the frequency of alpha and beta rhythms? Was monopolar monoarticular mounting used? Or was it an average montage?  

Electrode Placement and Montage: You've raised important questions regarding electrode placement and montage. EEG data were acquired and analyzed using a monopolar montage with ear reference.  

The frequency of the alpha band was analyzed in the occipital leads? Right or left? For the analysis of the alpha rhythm, data were recorded from bilateral occipital areas.  

Over which hemisphere were the scalpel electrodes placed? Electrodes were placed across the entire scalp to ensure comprehensive coverage.  

Was the interhemispheric asymmetry of the average frequency of the EEG rhythms analyzed by the authors or was it not in the pre-test and post-test? However, we did not examine the interhemispheric asymmetry in the average frequency of EEG rhythms in the pre-test and post-test.  

Was the beta rhythm frequency also analyzed in the occipital leads (above the occipital electrodes) or in the frontal leads? Right or left? Etc. The frequency of the beta rhythm was analyzed from the electrodes placed in the frontal areas of both cerebral hemispheres. 

For theta rhythm data were obtained from bilateral t5/t6 derivation on the side of the head, above the ear, in the temporal lobe region. 

In general, this section needs serious revision. I recommend the authors to present the results of statistical analysis of the characteristics of all the studied EEG frequency bands, including theta, alpha and beta (for examples, mean frequency, mean power, mean amplitude, coefficient of hemispheric asymmetry for all these characteristics) in the pre-test and post-test. 

Done 

Discussion: 

Lines 237-239 - the authors indicate that no positive EEG changes were found in the control group. Could this mean that "standard rehabilitation" is not effective? The results show a statistically significant improvement in the experimental group, while the control group maintains performance that is nearly unchanged and does not worsen.  

With what method of "standard rehabilitation" was the virtual reality test compared? Was "Standard rehabilitation" not prescribed to patients in the experimental group? 

The experimental group did VR while the control only paPER AND PENCIL APPROACH, AS BETTER SPECIFIED IN THE MANUSCRIPT (SEE ALSO TABLE 2) 

How does the virtual reality test method developed by the authors differ from similar numerous others previously known or actively studied? What is the novelty of this study? 

The novelty of this study consists in realizing a specific rehabilitative program, relating to right -lateralized cognitive abilities - using VRRS system, to treat the attention dysfunctions and memory alterations (visual and not), emerging after an ischemic stroke. WE USE eeg TO EVALUATE THIS treatment.  

Reviewer 2 Report

Comments and Suggestions for Authors

-It is not clear which band is actually enhanced due to VR which can be measureable through EEG.

- Why the authors have not included the use of other imaging technologies while explaining the importance of EEG such as fNiRS or fMRI.

- many figures are no legible. The quality is too bad to infer the idea of providing the results.

- Although demographics are provided, the mean and STD should be there to check the variations in the results.

- Whether the groups are balanced?

-Which statistical tools were applied to check the harness in among the groups.

 - Whether thr imaging based results are related with the behavioural results. Please clarify and add to the manuscript.

- References style are bad.

- Several papers of the Biomedecines journal need to be included to the references to show its relevence.

Author Response

Rev 2  

-It is not clear which band is actually enhanced due to VR which can be measureable through EEG. 

"The bands that have actually been enhanced are the Beta and Alpha bands" 

- Why the authors have not included the use of other imaging technologies while explaining the importance of EEG such as fNiRS or fMRI. The aim of the study was primarily to investigate the improvement of neurophysiological parameters. For future studies, we will consider a combined approach. 

- many figures are no legible. The quality is too bad to infer the idea of providing the results.                     "We have included new comprehensive figures in the text." 

- Although demographics are provided, the mean and STD should be there to check the variations in the results. We have calculated the means and standard deviations for the provided demographics and indices, as displayed in Table 2. This table presents a comprehensive summary of the key characteristics of both the experimental and control groups, including age, years of education, Barthel Index, Rankin Scale score, years from ischemic stroke, and sex distribution. The means and standard deviations offer a clear insight into the central tendencies and variability of the measured parameters within each group. 

- Whether the groups are balanced? Group Balance Assessment: To ensure the balance between the groups, we have conducted a thorough analysis of the provided demographic and clinical characteristics. As shown in Table 2, we have assessed the distribution of sex, age, years of education, Barthel Index, Rankin Scale score, and years from ischemic stroke between the two groups. The statistical tests indicate that there are no significant differences between the groups for any of these variables. Therefore, we can confidently conclude that the groups are well-balanced in terms of these important baseline characteristics. 

-Which statistical tools were applied to check the harness in among the groups. We utilized appropriate statistical tools for assessing the homogeneity among the groups. In particular, we performed independent samples t-tests for continuous variables and chi-squared tests for categorical variables to compare the two groups. The results are shown in Table 2, with p-values associated with each comparison 

 - Whether thr imaging based results are related with the behavioural results. Please clarify and add to the manuscript. 

- References style are bad. 

Reference style has been corrected 

- Several papers of the Biomedecines journal need to be included to the references to show its relevence. 

We added relevant papers of Biomedicines. 

Round 2

Reviewer 1 Report

Comments and Suggestions for Authors

The authors have made changes to the manuscript, but they have not taken into account all the comments of the reviewer.

Abstract:

Line 26 - The author again wrote that the sample size was 28 participants. I have doubts about the correctness of the presented results.

Line 27 - Is it the mean or the median? If this is the mean, then there is no standard deviation.

Materials and methods:

Line 177 - Please remove the abbreviation from the subsection name.

Line 222 - Indicate in which hemisphere you studied the alpha, beta and theta frequency bands? On the side of a stroke? Or not?

Results:

The sample size is very small, each subgroup included only 15 participants. I draw your attention to Table 3. It is incorrect to calculate percentages in this case.

Add the name of the first column.

Add units for each indicator you are analyzing (in the first column).

Line 256 - Add a name for the first column in Table 4. Add units of measurement for each frequency band in the first column (Hz). Add units for 2-3 and 5-6 columns (M±SE).

The authors answered my question that they calculated the average frequency of the studied EEG rhythms without taking into account interhemispheric asymmetry. Firstly, it is unclear to me how such a calculation was carried out. Secondly, the average value (excluding hemispheric asymmetry) is incorrect, because the patients suffered a stroke in the right hemisphere.

Figures 3-5 are uninformative. In addition, the power of the alpha rhythm and its zonal gradient worsened in patient 10 (experimental group) in the post-test. On the contrary, in the patient in the control group (Figure 5), the dynamics of the alpha rhythm in the post-test (compared with the pre-test) is much better (there was an increase in the power of the alpha rhythm in the parietal-occipital regions, an improvement in the zonal gradient). The zonal gradient of the beta frequency band in all cases is very similar to the zonal gradient of the alpha rhythm (this happens when the electrode helmet is located low on the back of the head, that is, not according to the standard).

This section is still the "weakest link" of the manuscript.

Although, this section should be the strongest and most informative in article.

Comments on the Quality of English Language

Authors need to improve the style of the English language.

Author Response

The authors have made changes to the manuscript, but they have not taken into account all the comments of the reviewer.

Abstract:

Line 26 - The author again wrote that the sample size was 28 participants. I have doubts about the correctness of the presented results.

This was an error, because we did not correct the abstract. Patients investigated were 30.

Line 27 - Is it the mean or the median? If this is the mean, then there is no standard deviation.

It is a mean, we have added SD in abstract session.

Materials and methods:

Line 177 - Please remove the abbreviation from the subsection name.

We have removed the abbreviation, as suggested.

Line 222 - Indicate in which hemisphere you studied the alpha, beta and theta frequency bands? On the side of a stroke? Or not?

We studied the alpha, beta and theta frequency bands on both hemispheres,  but we focused mainly on the right one.

Results

The sample size is very small, each subgroup included only 15 participants. I draw your attention to Table 3. It is incorrect to calculate percentages in this case.

We have eliminated the percentages calculated, as indicated. 

Add the name of the first column.

We have added the name of the first column.

Add units for each indicator you are analyzing (in the first column).

In the first column, we have added for each indicator.

Line 256 - Add a name for the first column in Table 4. Add units of measurement for each frequency band in the first column (Hz).

We have added the name for the first column in Table 4 with the units of measurement for each frequency band. Add units for 2-3 and 5-6 columns (M±SE).

We have added the units for 2-3 and 5-6 columns (M±SE).

The authors answered my question that they calculated the average frequency of the studied EEG rhythms without taking into account interhemispheric asymmetry. Firstly, it is unclear to me how such a calculation was carried out. Secondly, the average value (excluding hemispheric asymmetry) is incorrect, because the patients suffered a stroke in the right hemisphere.

In our study, we analyzed the bilateral EEG bands, but focusing on the right hemisphere, and the results were obtained by averaging epochs of data subjected to analysis. Then, we were not able to calculate the interhemispheric asymmetry, which was important as correctly concerned by the reviewer. Therefore we added it in the limitation.

Finally, although the present study focused on evaluating the power in the alpha, beta, and theta frequency bands in both hemispheres [70], we had paid attention on the affected right hemisphere, and we did not calculate interhemispheric asymmetry.  This latter would have definitely provided more detailed data on the functional connectivity (and therefore functional recovery) between the two cerebral hemispheres. However, the analysis of frequency band power is an approach also used in other studies to examine the intensity of brain waves in different regions or frequency bands. This analysis can be conducted on data from individual hemispheres or specific regions without considering the symmetry parameter between the two hemispheres."

In future research, we plan to integrate a comprehensive neurophysiological assessment of symmetry and coherence parameters along with hemodynamic studies using fNIRS to achieve improved spatiotemporal resolution”

Figures 3-5 are uninformative. In addition, the power of the alpha rhythm and its zonal gradient worsened in patient 10 (experimental group) in the post-test. On the contrary, in the patient in the control group (Figure 5), the dynamics of the alpha rhythm in the post-test (compared with the pre-test) is much better (there was an increase in the power of the alpha rhythm in the parietal-occipital regions, an improvement in the zonal gradient). The zonal gradient of the beta frequency band in all cases is very similar to the zonal gradient of the alpha rhythm (this happens when the electrode helmet is located low on the back of the head, that is, not according to the standard).

We are sorry for the bad image quality which makes it difficult to understand the results, as well as for the possible errors on bands interpretation. For this reason, we have proposed  a new clearer image focusing on a subject of the EG, to try overcoming the proper concerns of the reviewer.

This section is still the "weakest link" of the manuscript. Although, this section should be the strongest and most informative in article.

We have now improved this section, as suggested by the reviewer.

Reviewer 2 Report

Comments and Suggestions for Authors

The authors have removed all this reviewer's concerns.

Author Response

thanks for the positive evaluation

Round 3

Reviewer 1 Report

Comments and Suggestions for Authors

The authors modified the manuscript repeatedly. Thank you for this work.

Recommendations for technical correction:

Line 23, lines 27 - 36, 167-171, 225, 254-279, 348-394: remove the bold text selection.

Since this study has many limitations, I recommend the authors to single out the Limitations as an independent one.

Author Response

Bold has been removed, as suggested

A separete paragraph on limitation and future perspective has been added.